# Clinical Significance of the Duality of Wnt/β-Catenin Signaling in Human Hepatocellular Carcinoma

**DOI:** 10.3390/cancers14020444

**Published:** 2022-01-17

**Authors:** Tomoko Aoki, Naoshi Nishida, Masatoshi Kudo

**Affiliations:** Department of Gastroenterology and Hepatology, Kindai University Faculty of Medicine, 377-2 Ohno-Higashi, Osaka-Sayama 589-8511, Japan; tomoko.aoki@med.kinda.ac.jp (T.A.); m-kudo@med.kindai.ac.jp (M.K.)

**Keywords:** hepatocellular carcinoma, Wnt/β-catenin, HNF4α, FOXM1, Gd-EOB-DTPA-enhanced MRI, FDG-PET/CT

## Abstract

**Simple Summary:**

Wnt/β-catenin mutations in HCCs have a dual phenotype. Jekyll phenotype is defined by high *HNF4α* status which represses EMT, found in well-to-moderately differentiated HCCs with low AFP levels and good prognosis. *HNF4α* induces the expression of OATP1B3, which can be recognized as higher enhancement nodules in the hepatobiliary-phase of Gd-EOB-DTPA-enhanced MRI. On the other hand, Hyde phenotype is defined by high *FOXM1* status which promotes EMT, found in poorly differentiated HCCs with high AFP levels, positive stem cell markers and poor prognosis. *FOXM1* induces the expression of GLUT1, which can be recognized with FGD-PET/CT. HCC with activated Wnt/β-catenin signaling pathway. It could be hypothesized that the former should be mainly resistant to immune checkpoint inhibitors, while the latter would be immune activated and exhausted. In the future, the activation of the Wnt/β-catenin signaling pathway should be detected non-invasively and applied to the treatment of advanced HCC.

**Abstract:**

Combination therapy with immune checkpoint inhibitors (ICIs) and vascular endothelial growth factor inhibitors has been approved as a first-line treatment for unresectable hepatocellular carcinoma (HCC), indicating a critical role of ICIs in the treatment of HCC. However, 20% of patients do not respond effectively to ICIs; mutations in the activation of the Wnt/β-catenin pathway are known to contribute to primary resistance to ICIs. From this point of view, non-invasive detection of Wnt/β-catenin activation should be informative for the management of advanced HCC. Wnt/β-catenin mutations in HCC have a dual aspect, which results in two distinct tumor phenotypes. HCC with minimal vascular invasion, metastasis, and good prognosis is named the “Jekyll phenotype”, while the poorly differentiated HCC subset with frequent vascular invasion and metastasis, cancer stem cell features, and high serum Alpha fetoprotein levels, is named the “Hyde phenotype”. To differentiate these two HCC phenotypes, a combination of the hepatobiliary phase of gadolinium-ethoxybenzyl-diethylenetriamine (Gd-EOB-DTPA)-enhanced magnetic resonance imaging and fluoro-2-deoxy-D-glucose-PET/CT may be useful. The former is applicable for the detection of the Jekyll phenotype, as nodules present higher enhancement on the hepatobiliary phase, while the latter is likely to be informative for the detection of the Hyde phenotype by showing an increased glucose uptake.

## 1. Introduction

Cancer is often explained by new applications in cell-to-cell evolutionary biology. Neoplasms grow in a complex ecosystem, and natural selection is a powerful defense mechanism in cancer. The development of cancer is attributed to clonal selection; genetic mutations that favor cell proliferation are called driver mutations, and malignant neoplasms emerge by acquiring a growth advantage through cell proliferation [1]. It has been reported that three or more driver mutations are required for the emergence of malignant tumors [2].

Hepatocellular carcinoma (HCC) is a heterogeneous tumor, both in the pathological and molecular aspects. Critical genomic hits at the preneoplastic stages are not well defined, although aberrant methylation, genomic instability, and telomerase reverse transcriptase (*TERT*) activation have been reported [3]. In overt HCC, deregulated oncogenes, such as *MET*, catenin B1 (*CTNNB1*), *MYC*, and c-terminal cyclin D1 (*CCND1*), and tumor suppressor genes, including *TP53*, phosphatase, tensin homolog deleted on chromosome 10 (*PTEN*), cyclin-dependent kinase inhibitor 2A (*CDKN2A*), and cadherin 1 (*CDH1*), have been characterized through genetic and epigenetic analyses [3]. The mutation profile of HCC is heterogeneous among nodules, even in the same liver, suggesting complex variations in the oncogenic pathway leading to hepatocarcinogenesis. For example, mutations in the *TERT* promoter are found in 60–70% of HCCs, *TP53* in 21–35%, *CTNNB1* in 16–40%, *AXIN1* in 5–15%, adenomatous polyposis coli (*APC*) in 2–3%, phosphatidylinositol-4,5-bisphosphate 3-kinase catalytic subunit alpha (*PIK3CA*) in 1.5%, rat sarcoma (*RAS*) in 1.3%, and *PTEN* in 1% [4,5]. Mutations in *CTNNB1* and *TP53* are mutually exclusive and are often defined as two distinct tumor phenotypes in HCC [4,6]. On the other hand, mutations in *AXIN1* and *TP53* can occur together and are defined as the G2 subclass of Boyault [4].

The extremely poor prognosis of patients with HCC is mainly due to the high frequency of early tumor metastasis, which denies the opportunity for surgical resection. Multi-organ metastasis or vascular invasion are commonly observed in HCC with epithelial-to-mesenchymal transition (EMT), which is one of the characteristics of malignant neoplasms induced by the dysregulation of the Wnt signaling pathway. 

Attempts to predict the activation of the Wnt signaling pathway by gadolinium-ethoxybenzyl-diethylenetriamine (Gd-EOB-DTPA)-enhanced magnetic resonance imaging (MRI) have been reported, and higher enhancement nodules in the hepatobiliary phase have been shown to express the organic anion transporting polypeptide 1B3 (OATP1B3), which can be an indicator of Wnt signal activation [7,8,9,10]. The majority of HCCs with Wnt/β-catenin activation and OATP1B3 expression have been reported to have a good prognosis with reduced vascular invasion and distant metastases [7]. 

The following two distinct phenotypes of HCC with activated Wnt signaling have been identified: the “Hyde phenotype”, associated with EMT, and the non-proliferative “Jekyll phenotype”. These have been frequently pointed out, but their molecular backgrounds are not yet fully understood. Since mutation analysis of HCC is often performed on resected specimens, there is a lack of studies on the Hyde phenotype because of selection bias, where patients with less aggressive tumors are prone to undergo resection. It is conceivable that several molecules located downstream of the Wnt signaling pathway are involved in the acquisition of stemness and EMT phenotypes. In this review, we focus on the role of the Wnt target and downstream molecules involved in the conflicting phenotypes of HCC with Wnt/β-catenin activation and discuss their clinical implications.

## 2. Jekyll and Hyde Phenotype of HCC

As described above, activation of the Wnt/β-catenin pathway may induce different phenotypes of HCC, that is, progressive (Hyde phenotype) and non-progressive (Jekyll phenotype). The Hyde phenotype is defined by cancer stem cell features, high serum alpha-fetoprotein (AFP) levels, vascular invasion, and multi-organ metastasis, where EMT and angiogenesis may occur. The Jekyll phenotype is defined as a subset with good prognosis with localized, low recurrence rate, and lack of high serum level tumor markers. Wnt/β-catenin signaling participates in the establishment of EMT in a variety of cancers.

## 3. Epithelial-to-Mesenchymal Transition in the Cancer Microenvironment

Previous studies have revealed that the induction of EMT is required for body planning and differentiation of multiple tissues and organs, but also plays an important role in the progression of cancer cells. Phenotypic changes in fibroblasts are an initial step in the successful metastasis of cancer cells. Cancer-associated fibroblasts induce EMT via the transforming growth factor beta (TGF-β)/interleukin (IL)-6 signaling pathway [11] (Figure 1). Cancer cells acquire invasiveness and metastatic potential through drastic changes in the composition of the cytoskeleton and the acquisition of motility by the same mechanisms observed in embryos. EMT may also promote the growth of metastatic cancer cells at other sites. The core features of EMT are the downregulation of E-cadherin, which is essential for cell–cell adhesion, and the upregulation of vimentin, which represents the mesenchymal phenotype [12].

EMT also contributes to tissue repair, but it adversely induces fibrosis and cancer progression through multiple signaling pathways, such as TGF-β, Wnt/β-catenin, Notch, hepatocyte growth factor (HGF), epidermal growth factor (EGF), fibroblast growth factor (FGF), and the hypoxia-inducible factor (HIF), which was reported to trigger EMT [13]. These signaling pathways activate a group of EMT-related transcription factors such as Twist, the Snail family (Snail (*Snai1*), Slug (*Snai2*), Smuc), the zinc finger E-box binding homeobox (ZEB) family (EF1/ZEB1, SIP1/ZEB2), and E12/E47, and EMT-related transcription factors directly or indirectly suppress E-cadherin production. Previous studies have shown that Wnt/β-catenin signaling is involved in EMT in many cancers; however, because of differences in cellular environment and tissue specificity, the phenotypes and events of downstream molecular are slightly different. Among the EMT-related transcription factors, *Snail1* and *Snail2* were shown to be directly regulated by Wnt/β-catenin [13]. Snail accelerates cancer invasion by upregulating matrix metalloproteinase (MMP) expression and is associated with a poor prognosis of HCC [14]. Twist is also indirectly activated by Wnt/β-catenin signaling, and accumulation of Twist protein by β-transducin repeat-containing E3 ubiquitin protein ligase (β-TRCP) can promote tumor cell motility and cancer metastasis [13]. In HCC, Snail and Twist have been shown to be important in the process of metastasis, and these transcription factors are often induced by TGF-β [15]. The relationship between fibroblasts, EMTs, and cancer stem cells has been well investigated in scirrhous HCC, in which tumors are abundant in the fibrous portion; Seok et al. suggested that the fibrous stromal component in HCC might contribute to the acquisition of EMT-related genes and overexpression of TGF-β signaling in HCC [11].

## 4. Wnt Signaling Pathway and Related Molecules

Wnt signaling is involved in almost all aspects of embryonic development and regulates homeostatic self-renewal in many adult tissues. The Wnt gene encodes small, secreted proteins that are preserved in all animal genomes, and genetic or epigenetic alterations of the Wnt gene have been reported in cancer. The molecular weight of Wnt genes is approximately 40,000, and 19 types of Wnt genes have been identified in mammals. The following three pathways are believed to be involved in the downstream Wnt receptor: the canonical Wnt/β-catenin pathway, the non-canonical planar cell polarity pathway, and the non-canonical Ca^2+^ pathway. Of these three, the canonical pathway is best understood—see Clevers’ review for details [16]. The Frizzled proteins are seven-pass transmembrane receptors with a cysteine-rich domain at its extracellular N-terminal. The lipoprotein receptor-related protein (LRP) 5, and LRP6 are single-pass transmembrane molecule. 

### 4.1. Canonical Pathway of Wnt/Β-Catenin Signaling

In the canonical pathway, the trimeric complexes formed by Frizzled, LRP5, and LRP6 need to be expressed on the surface. Activation of the canonical Wnt signaling pathway is initiated by the binding of transforming ligands such as Wnt3a, Wnt1, and Wnt7 to the Frizzled/LRP co-receptor complex (Figure 2) [16]. 

In the absence of Wnt, Axin functions as a scaffolding protein of the β-catenin pathway, specifically and efficiently promoting the phosphorylation of β-catenin by glycogen synthase kinase 3β (GSK-3β). Intracytoplasmic β-catenin binds to the central part of Axin, and APC binds to the regulator of the G protein signaling (RGS) domain, forming the Axin/APC complex. In the Axin/APC complex, multiple β-catenins are sequentially phosphorylated by GSK-3β, and phosphorylated β-catenins are successively degraded by the ubiquitin-proteasome system. Phosphorylation of Axin and APC enhances the phosphorylation of β-catenin (Figure 2). 

Wnt binding to the Frizzled/LRP trimeric co-receptors enhances the phosphorylation of the cytoplasmic tail of LRP5/6. These phosphorylation signals suppress the phosphorylation of β-catenin in a GSK-3β-dependent manner. The hypo-phosphorylated β-catenin escapes proteasome-mediated degradation and accumulates in the cytoplasm. The recruitment of Axin away from the destruction complex leads to stabilization and accumulation of β-catenin. Caveolin-dependent endocytosis of the Axin complex by the plasma membrane LRP6 receptor upon stimulation by Wnt3a is considered essential for β-catenin stabilization. In the nucleus, β-catenin binds to T-cell factor/lymphoid enhancing factor 1 (TCF/LEF1) to promote the transcription of Wnt target genes (Figure 2). R-spondin enhances Wnt/β-catenin signaling by increasing the expression of Frizzled receptors. Loss of components of the Wnt canonical pathway can affect a variety of tissues and produce drastic phenotypes [16].

β-catenin, whose phosphorylation and degradation are blocked by Wnt canonical stimulation, enters the nucleus and binds to the TCF/LEF1 family of transcription factors to regulate the expression of target genes. In the absence of Wnt/β-catenin signaling, TCF acts as a repressor of Wnt/β-catenin target genes by forming a complex with the Groucho/TLE transcriptional co-repressor. Wnt signaling induces conflicting events, such as cell proliferation and terminal differentiation [16]. In other words, the major role of Wnt signaling is to ultimately activate the transcriptional program, and there is no intrinsic restriction in the types of biological events that may be regulated downstream of Wnt signaling [16].

### 4.2. Wnt Target Gene

The Wnt/β-catenin signaling pathway is crucial for both normal development and carcinogenesis [17]. More than 100 different target genes of Wnt signaling have been reported, but they can be roughly classified into three types as follows [18]: first, genes related to cell proliferation and cell cycle, such as *c-**myc* and *cyclin D1*; second, genes related to body axis formation and organogenesis; third, genes related to the regulation of Wnt signaling, such as *Wnt3a*, *Axin2*, *Frizzled*, *LEF1*, *TCF1*, and *LGR5*.

The transcription factors Snail (Snail1), Slug (Snail2), and Twist are the key inducers of EMT that are associated with tumor metastasis, and *Snail1*, *Snail2*, and *Twist* are well-known Wnt target genes. Interactions between β-catenin and TGF-β signaling pathways mediate EMT [19] (Figure 1). It has been reported that *Snail* and *Twist* are also targets of TGF-β signaling, which is one of the critical pathways of HCC formation [20], and Yang has revealed that overexpression of Snail and/or Twist correlates with the poor prognosis of HCC [15]. Other Wnt target genes include those of *MMPs* and *VEGF*, which are known to be involved in neovascularization and interact with cancer stem cell markers such as Sal-like protein 4 (SALL4), epithelial cell adhesion molecule (EpCAM), and the adhesion molecule E-cadherin.

## 5. Canonical Wnt/β-Catenin Mutation and Carcinogenesis

The canonical β-catenin pathway regulates the cell cycle and cell proliferation through gene expression, including *cyclin D1* and *c-myc*. Therefore, genetic mutations in the regulatory components of the β-catenin pathway disrupt cell cycle control, resulting in carcinogenesis. Through a comprehensive analysis of cancer genomes based on deep sequencing, it became evident that mutations in *APC*, *CTNNB1*, and *Axin* are frequently detected in many types of human cancers, such as colorectal and liver cancers, with β-catenin accumulation [17]. APC may also be inactivated through epigenetic mechanisms, and dense methylation of APC has been reported in HCC [6]. The frequency of Wnt/β-catenin mutations is reported to be highest in colorectal cancer (>90%), followed by ovarian cancer (~50%), hepatocellular carcinoma (~40%), and malignant melanoma (~30%) [18].

## 6. Dual Aspect of Wnt/β-Catenin Mutation 

Aberrantly activated Wnt/β-catenin signaling contributes to increased proliferation, loss of epithelial morphology, and de-differentiation. Wnt targets the EMT-related transcription factors Snail and Twist, which promote metastasis and invasion in carcinogenesis and contribute to poor prognosis [19]. However, there are conflicting reports on the relationship between Wnt signaling activation and prognosis. Although canonical Wnt/β-catenin signaling thought to contribute to cancer development, its role in cancer progression remains controversial.

In malignant melanoma, several clinical studies have shown that overall survival rate is increased in patients with accumulation of nuclear β-catenin [21]. However, in a mouse melanoma model with a mutant *BRAF* and *PTEN* genotype, β-catenin signaling activation leads to promoted tumor development and promotion of metastasis [22]. Webster showed the involvement of Wnt5a and alternative reading frame (ARF) 6 in promoting β-catenin expression and tumor metastasis [23]. A study involving colorectal cancer after recurrence showed that the overall survival rate was higher in molecular subtypes with high expression of Wnt target genes compared to those with low expression of these genes [24]. In breast cancer, it has been reported that Wnt signaling is activated in more than 50% of patients, and Wnt/β-catenin and cyclin D1, which is one of the targets of this pathway, are negatively associated with overall survival [25].

Aberrant activation of canonical Wnt/β-catenin signaling has been reported in approximately 40% of HCC cases [5]. In contrast, a subgroup of HCCs with an enhanced non-canonical Wnt/TGF-β axis has also been reported [26]. The uniform features of canonical *CTNNB1* mutated HCCs have been reported as follows: well-differentiated, cholestatic, with microtrabecular and pseudoglandular patterns, and without inflammatory infiltrates [4,7,8]. These studies included mainly surgically resected HCCs, and it is important to note that these results were attributed to a biased cohort that did not include advanced HCC showing EMT. As will be discussed in detail later, HNF4α has been associated with low-grade HCC without inducing EMT in this subgroup.

Reports of Wnt/β-catenin mutations in poorly differentiated HCCs are very limited. Inagawa et al. reported the nuclear accumulation of β-catenin and prognosis in solitary HCCs (including 24% poorly differentiated types) smaller than 30 mm. The results showed that in well-differentiated HCC, intranuclear accumulation of β-catenin was negatively correlated with mortality outcome, while in poorly differentiated HCC, intranuclear accumulation of β-catenin was a significant poor prognostic factor [27]. Authors also reported a significant correlation between the nuclear accumulation of β-catenin and decreased expression of E-cadherin in the plasma membrane [27]. E-cadherin is a key cell adhesion protein implicated in tumor invasion or human carcinoma suppression and a binding partner of β-catenin. Although there are various reports on the behavior of E-cadherin and β-catenin, decreased E-cadherin expression correlates with nuclear accumulation of β-catenin and has been reported to be a risk factor for recurrence after surgical treatment. On the other hand, Yamashita indicated that EpCAM is a Wnt/β-catenin signaling target gene [28] and EpCAM-positive HCC cells exhibit activation of the Wnt/β-catenin signaling pathway, epithelial cell morphology, and high tumorigenicity, resulting in poor prognosis [29,30].

## 7. Downstream Wnt/β-Catenin Master Genes Define the HCC Dual Phenotype 

Reports of canonical Wnt/β-catenin signaling activation and the malignant phenotype or prognosis of HCC have been confounded by this duality. Yamashita et al. reported that this paradox may be explained by the exclusive expression and activation of hepatocyte nuclear factor (HNF)4α and forkhead box M1 (FOXM1) [31] (Figure 1). β-catenin and glutamine synthetase (GS) are highly expressed in well-differentiated to moderately differentiated HCC, and are, to some extent, less frequently expressed in poorly differentiated HCC (Figure 3). Expression of HNF4α, which identifies genes detected in mature hepatocytes, is found predominantly in less aggressive HCC, while FOXM1 oncogene expression has been reported to be associated with poorly differentiated, stem cell marker-positive HCC [31] (Figure 3).

### 7.1. Jekyll Phenotype Defined by HGF4α

HNF plays an important role in the development of the liver. HNF4 is a member of the steroid hormone receptor superfamily, and HNF4 messenger RNA (mRNA) has been shown to be present in the kidney, intestine, and abundantly in the liver, but is absent in other tissues [32]. HNF4α was first identified as an important regulator of genes involved in hepatocyte differentiation and cell adhesion [32]. HNF4α is also known to repress several genes involved in hepatocyte proliferation, such as cyclin D1. There is also evidence to suggest that HNF4α expression may be defective in human HCC.

Among the hepatic transcription genes identified in previous studies on the transcriptional regulatory network in hepatocytes, HNF4α was also identified as one of the most important hepatic transcription factors. HNF4α is probably the most upstream mediator, acting as a master gene that regulates the transcription cascade to promote differentiation, proliferation, and morphogenesis of the hepatic lineage. It has been reported that Wnt/HNF4a interacts with LEF1 to affect liver zonation [33].

HNF4α has also been reported to regulate EMT in HCC cells by inhibiting Wnt/β-catenin transcription and enhancing the localization of β-catenin in the cell membrane [34]. Through a negative regulation mechanism of HNF4α on EMT, HNF4α has been found to be more common in well-differentiated HCC with less vascular invasion and metastasis and low AFP levels. Yang et al. found that HNF4α induces a negative feedback loop for Wnt/β-catenin and negatively regulates the expression of Snail and Slug [34] (Figure 3).

In a recent study, two HNF4α promoters were identified (P1 and P2), each of which expresses HNF4α transcriptional variants that are differentially expressed in human HCC and colon cancer [35]. Promoter P1 HNF4α (P1-HNF4α) is the isoform predominantly expressed in the adult liver and represses the expression of tumor-promoting genes such as c-myc, cyclins, and EMT-related genes in a circadian manner [35]. On the other hand, abnormal expression of P2-HNF4α may be involved in the development of HCC and provide strong transcriptional repression of the circadian clock genes, including the aryl hydrocarbon receptor nuclear translocator-like (*ARNTL*, also known as *BMAL1*) gene [35]. In normal hepatocytes, the P1-HNF4α isoform and the circadian protein BMAL1 are concomitantly expressed. HNF4α is also a key regulator of miR-122 expression in hepatocytes, which acts as a tumor suppressor and represses HCC development [36]. MiR-122 transiently suppresses both BMAIL1 and Clock, core circadian regulators. ADAM17 is a target of miR-122 and plays a critical role in liver tumorigenesis and progression in HCC [36].

### 7.2. Imaging Biomarker of Jekyll Phenotype

It has been well described that green hepatoma, characterized by tumor cholestasis and showing a high-intensity nodule on Gd-EOB-DTPA-enhanced-MRI, generally shows a good prognosis and carries CTNNB1 mutations. OATP1B3, a transporter of bile acids, was found to be highly expressed in green hepatoma, and HCCs with high expression of OATP1B3 were negative for cancer stem cell markers, including cytokeratin-19 (CK19) and EpCAM (Figure 4).

Gd-EOB-DTPA-enhanced MRI is a commonly used modality for the diagnosis of HCC. The liver-specific contrast agent used in MRI imaging maintains high diagnostic accuracy because the contrast agent is taken up by the hepatocytes via OATP1B3 expressed on the vascular side, indicating the potential of EOB-MRI as an imaging biomarker for predicting CTNNB1 mutations. 

Some studies indicated that the high expression of OATP1B3 was strongly correlated with a higher enhancement in hepatobiliary phase of Gd-EOB-DTPA-enhanced MRI [7,8]. The study of immunostaining and RNA sequencing of human HCC cells revealed that the activation of Wnt/β-catenin signaling induced the expression of OATP1B3, indicating that both activations are closely related [8]. Regarding the prediction of Wnt/β-catenin activation using Gd-EOB-DTPA-enhanced MRI, the sensitivity, specificity and accuracy were 78.9%, 81.7%, and 81.2%, respectively, in well-to-moderately differentiated HCC, with the relative enhancement ratio (RER) cut-off point of 0.90 [8]. This study did not show that OATP1B3 is directly regulated by Wnt/β-catenin. On the other hand, it has been reported that co-activation of Wnt/β-catenin and HNF4α is an important regulator of the OATP1B3 enhanced expression in HCC tissue cells, leading to an increased hepatobiliary phase in Gd-EOB-DTPA-enhanced MRI [37]. Based on this knowledge, the non-invasive detection of Wnt/β-catenin-activated HCC nodules using Gd-EOB-DTPA-enhanced MRI to predict their resistance to ICI treatment in a previous study of ours [10].

### 7.3. Hyde Phenotype Defined by FOX Transcription Factors

The vital role of the Wnt/β-catenin signaling pathway in the proliferation and self-renewal of normal cells and cancer stem cells is commonly accepted. Among HCCs with Wnt/β-catenin activation, some show an aggressive phenotype that is positive for EMT transcription factors and cancer stem cell markers, including CK19, EpCAM, and SALL4 [27,31]. In addition to stem cell markers, many poorly differentiated HCCs have been reported to have elevated Ki67 levels [29]. This aggressive subtype was reportedly associated with activation of the transcription factor FOXM1 [31,38,39,40], downstream of the Wnt/β-catenin signaling pathway [41]. 

FOX proteins are transcription factors that regulate embryonic development and tissue homeostasis, and they are highly conserved across species. There are 44 FOX family members in humans, which are subclassified into 19 classes (A to S) based on sequence resemblance of the DNA-binding forkhead domain [42]. Functional studies of FOX have revealed that transcriptional activity differs markedly even when the domain sequences are similar. While some FOX transcription factors are ubiquitously expressed, almost all family members demonstrate strongly restricted spatial or temporal expression patterns. About half of the mammalian FOX transcription factors have been found to be involved in the Wnt signaling pathway, primarily in cancer cells. Evidence suggests that this close association between FOX transcription factors and the Wnt pathway contributes to cancer initiation and progression. The FOX family includes both activators and inhibitors of the Wnt pathway [42]. The Wnt pathway activators FOXM1 and FOXG1 have been reported to be expressed in HCC. FOXM1 is expressed in many cancers and has been demonstrated to be directly regulated by the canonical Wnt/β-catenin signaling pathway, contributing to the development of the cancer phenotype, including proliferation, decreased apoptosis, metastasis, and drug resistance [41]. Overexpression of FOXG1 is associated with resistance to apoptosis and metastasis in various cancers, including HCC.

The master control gene *FOXM1* codifies for a proliferation-specific transcription factor that plays a critical role in cell cycle progression and initiation of cancer development. Deletion of *FOXM1* in HCC inhibited tumor growth, indicating that FOXM1 is required for tumor progression [43]. It has been reported that FOXM1 strongly enhances the nuclear translocation of β-catenin in normal cells and cancer cells, and nuclear shuttling of β-catenin by FOXM1 is commonly observed in many cancers such as glioblastoma, meningiomas, and pancreatic ductal adenocarcinoma [41]. Besides its role in β-catenin shuttling, FOXM1 also enhances the expression of Wnt target genes at the chromatin level. Wnt3a increases the level and nuclear translocation of FOXM1, which binds directly to β-catenin and enhances the nuclear localization and transcriptional activity of β-catenin. FOXM1 is known to be a component of the complex that promotes the transcription of Wnt target genes, *c-**myc* and *cyclin D1*. Cyclin D1-positive HCC has been reported to show an aggressive tumor phenotype and a poor prognosis, and it is reasonable to assume that the Wnt/FOXM1 axis promotes an aggressive phenotype of this subset [44] (Figure 3).

Continuous daily injections of a cell-penetrating ARF peptides in HCCs in mice with FOXM1 activity has also been shown to inhibit tumor cell proliferation and angiogenesis and significantly increase apoptosis within the HCC region [43]. P19 ARF (p14ARF in humans) exerts tumor suppressive effects in both p53-dependent and p53-independent manners [43]. The p14*ARF* tumor suppressor gene has been reported to play an important role in the pathogenesis of HCC, showing genetic alteration, hypermethylation of the promoter, and loss of heterozygosity in the 9p21 region [45]. Other studies have shown that overexpression of FOXM1 strongly correlates with disease progression and poor prognosis in many human malignancies. FOXM1 is often reported to be an indicator of aggressive progression and poor prognosis in HCC [31,38,39,40]. It is also known that FOXM1 strongly correlates with cancer stem cell features and EMT characteristics [39]. FOXM1 also promotes transcription of STMN1, which enhances cell motility, and stimulates the expression of lysyl oxidase (LOX) and lysyl oxidase-like 2 (LOXL2), which are involved in the formation of pre-metastatic niches in organs targeted for distant metastasis. In HCC cells, FOXM1 increases the expression of VEGF, which is required for angiogenesis, and contributes to the acquisition of an aggressive tumor phenotype [46].

### 7.4. Glucose Metabolism and Warburg Effect

Intracellular energy metabolism is highly conserved in many organisms. Inhibition of energy metabolism pathways are involved in gene expression through epigenetic mechanisms. Various proto-oncogenes induce the expression of glucose transporters and glycolytic enzymes that lead to the elevation of the glycolytic system, indicating that tumor cells are highly dependent on the glycolytic system [47]. In addition, the tumor microenvironment is extremely low in glucose compared to other tissues because tumor cells consume large amounts of glucose [48]. The local environment of the tumor is hypoxic, low pH due to high lactate, and low nutrition. The Warburg effect is that tumor cells rely on the glycolytic system for energy even under aerobic oxygenation, taking up glucose and metabolizing it to lactate [49]. Metabolic reprogramming allows HCC cells to adapt and survive in a nutrient-limited microenvironment in a glucose-dependent manner and high expression of glucose transporter 1 (GLUT1) in HCC was significantly correlated with poor survival [50]. In addition, it has been reported that the anti-apoptotic protein poly polymerase (PARP) 14 promotes the Warburg effect and high expression of PARP14 in HCC cell was associated with poor prognosis [51].

### 7.5. Imaging Biomarker of Hyde Phenotype

To date, HNF4α-negative and FOXM1-positive HCC with activated Wnt/β-catenin signaling, namely the aggressive phenotype, should be a candidate for systemic chemotherapy, including ICIs, but has not yet been effectively diagnosed using current imaging techniques. The promoter of FOXM1 contains a STAT3 binding site, and FOXM1 has been reported to regulate the transcription of GLUT1 through the *SLC2A1-AS1/STAT3* interaction [52] (Figure 4). STAT3 knockdown significantly decreased the expression of FOXM1 and GLUT1 at the transcriptional and translational levels, and significantly reduced the promoter activity of FOXM1 [52]. In addition, with FDG-PET/CT imaging, Ahn et al. identified an SUV signature with high accumulation in breast cancer, and verified a subset showing the same expression signature in the Cancer Genome Atlas database with FOXM1 activation [53]. Based on this knowledge, it is conceivable that the Wnt/β-catenin-positive poor prognosis group (the Hyde phenotype) should be diagnosticated with FDG-PET/CT, while in the less aggressive Jekyll HCC phenotype EOB-MRI could be an ideal imaging biomarker (Figure 4).

## 8. Compatibility with Previous Molecular Classifications of HCC and Development of Anticancer Drugs

The first-line therapy for unresectable HCC with Child-Pugh class A liver function is a combination of ICIs and VEGF inhibitors [54,55]. However, 20% of patients do not respond effectively to ICIs, and mutations in the activation of the Wnt/β-catenin signaling pathway are known to be the cause of primary refractoriness to ICIs [9].

Activation of canonical Wnt/β-catenin signaling, whose ligands are Wnt3a, Wnt1, and Wnt7, leads to two different phenotypes of HCC. One is attributed to the induction of HNF4α, which causes β-catenin activation in certain Hoshida S3 HCCs (Figure 5). In this subclass, the Wnt/β-catenin signal induces immune exclusion by reducing the recruitment of CD103+ dendritic cells, with the induction of ATF3 and downregulation of (c-c motif) ligand (CCL)4/5, CXCL1 and CCL20 [5,9]. This condition is referred to as “immune excluded” or “immune cold” subclass. Although HCCs are heterogeneous tumors, it has been reported that anti-PD-1/PD-L1 antibodies lose their efficacy when even one of the multiple nodules has Wnt/β-catenin mutations and HNF4α expression [10]. 

In contrast, activation of FOXM1 by the canonical Wnt/β-catenin signaling pathway may result in Hoshida S2 HCC, which shows high AFP and stem cell features [29,31] (Figure 5). The macrotrabecular/compact pattern, a pathological feature of the Hoshida S2 subclass, was associated with the activation of the oncogenes for YAP/TAZ and stemness markers EpCAM, CK19, and SALL4. The immune status of this subclass has been reported to elevate CCL5, abundant TIL, immunosuppressive receptors expression, enhanced IFNγ signaling and strong antigen presentation [56]. We believe that this subclass corresponds exactly to the Hyde phenotype, which is an immune microenvironment with high affinity for ICI treatment. The FOXM1 and β-catenin complexes promote c-myc transcription. Some HCCs expressing c-myc have been reported to escape the immune system by upregulating the canonical Wnt/β-catenin pathway [57]. The migration of immature dendritic cells through the basement membrane is dependent on MMP-9, where MMP-9 is regulated by Wnt and FOXM1. The stem cell-like Hoshida S2 HCC shows a high propensity for vascular invasion and a poor prognosis, but it has been reported to respond well to Wnt signaling inhibitors [28]. In addition, SALL4 forms a chromatin remodeling complex with histone deacetylase (HDAC), which has been reported to decrease the expression of SALL4 and the number of EpCAM-positive stem cells. Therefore, HDAC inhibitors may also be effective in treating HCC with stem cell features. A recently developed FOXM1 inhibitor reduced β-catenin abundance and activity in different cancer cell lines and attenuated tumor growth in mouse xenograft models.

In the non-canonical Wnt signaling pathway, binding of the weak trans-forming ligands Wnt5a, Wnt6, and Wnt11 promotes activation of the Wnt/TGF-β axis. The non-canonical Wnt/TGF-β signaling pathway is associated with the S1 subclass of Hoshida’s classification [26]. This subset shows T-cell infiltration into the immune exhaustion phenotype with ICI resistance [5] (Figure 5).

In a study of resected specimens, Wnt/β-catenin-positive HCC was reported to be around 30–40%, and OATP1B3-positive HCC recognized by the hepatobiliary phase of Gd-EOB-DTPA-enhanced MRI was reported to be around 15–20%, suggesting the possibility that a certain number of Wnt/β-catenin-positive, HNF4α-negative, FOXM1-positive HCC subsets exist. However, the study using resected specimens has a selection bias, in that patients with vascular invasion and multi-organ metastasis cannot not be subjected to surgical resection. The next challenge is to validate our findings in a variety of advanced HCC samples.

## 9. Conclusions

*HNF4α* is a tumor suppressor gene that represses EMT, which is predominantly found in well-to-moderately differentiated HCCs and defines a subset with low AFP levels and good prognosis. It is a target gene of the Wnt/β-catenin pathway and induces the expression of OATP1B3 by co-expression, which can be recognized as a non-invasive nodule with high enhancement in the hepatobiliary phase of Gd-EOB-DTPA-enhanced MRI.

On the other hand, HCCs with FOXM1 expression, located downstream of the Wnt/β-catenin pathway, are an aggressive and poor prognostic subset presenting high AFP levels, positive stem cell markers, poorly differentiated with EMT, and reduced E-cadherin expression. This subset lacks HNF4α expression and is not recognized as a nodule with higher enhancement by the hepatobiliary phase of Gd-EOB-DTPA-enhanced MRI. However, since it expresses GLUT1, it can be noninvasively recognized with FGD-PET/CT.

It has been reported that HCC with an activated Wnt/β-catenin signaling pathway is primarily resistant to first-line immune checkpoint inhibitors. In the future, the activation of the Wnt/β-catenin signaling pathway should be accurately detected and applied to the treatment of advanced HCC for which biopsy specimens cannot be obtained.

## Figures and Tables

**Figure 1 cancers-14-00444-f001:**
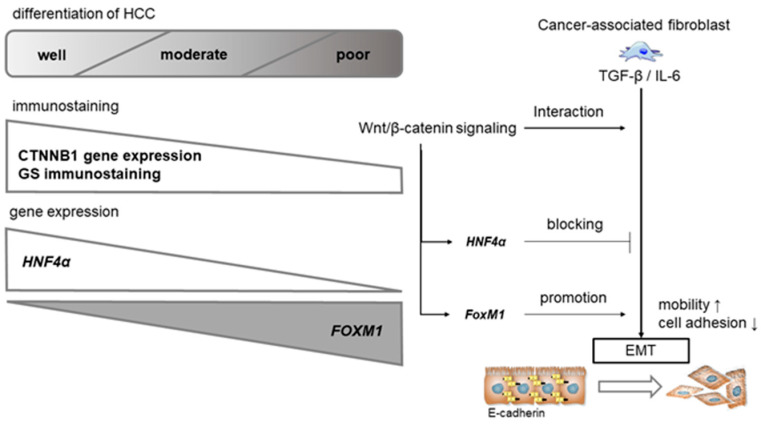
Wnt/β-catenin canonical pathway interacting with EMT and the TGF-β/IL-6 axis. It is well known that the TGF-β/IL-6 axis regulates EMT, and Wnt/β-catenin has been reported to directly promote EMT. Wnt/β-catenin activation mutations are present in approximately 40% of HCCs, especially in well-to-moderately differentiated HCCs. HNF4α, a target gene of Wnt, suppresses EMT, resulting in a non-aggressive phenotype with less metastasis and invasion. FOXM1, which is regulated by Wnt, promotes EMT and decreases E-cadherin expression, resulting in a poor prognostic phenotype with aggressive metastasis and invasion.

**Figure 2 cancers-14-00444-f002:**
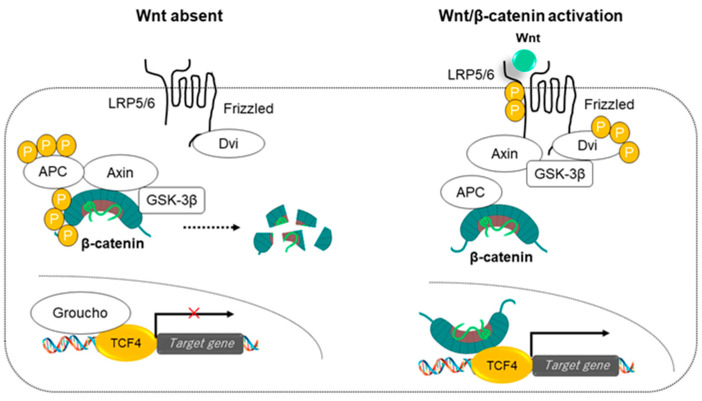
Canonical Wnt signaling pathway. (**Left panel**) When Wnt is not bound to the Frizzled/LRP coreceptor complex, GSK-3β phosphorylates β-catenin. The phosphorylated β-catenin is rapidly destroyed by the proteasome. In the nucleus, the binding of Groucho to TCF inhibits the transcription of Wnt target genes. (**Right panel**) Receptor complexes bound to Wnt activate canonical signaling pathways; Wnts induce phosphorylation of LRP by GSK-3β and stabilize β-catenin when Axin is recruited. In the nucleus, β-catenin displaces Groucho from TCF/LEF to promote the transcription of Wnt target genes.

**Figure 3 cancers-14-00444-f003:**
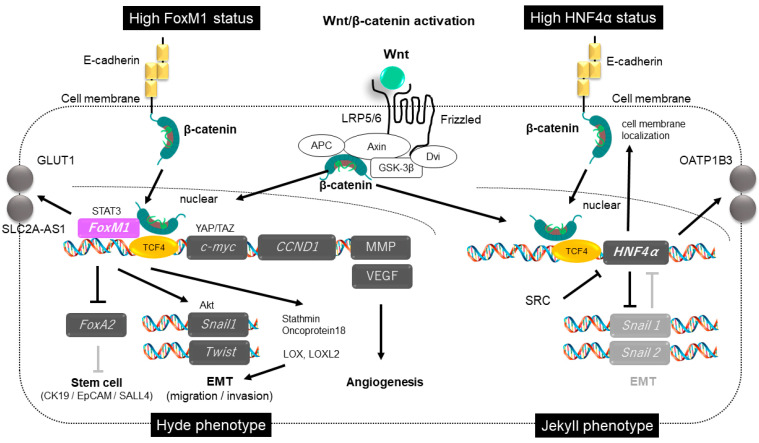
Dual aspects of Wnt/β-catenin: the Jekyll and Hyde phenotype. (**Left panel**) HCC cells with high *FoxM1* status. *FOXM1* strongly enhances nuclear translocation of β-catenin in HCC cells. Nuclear shuttling of β-catenin by *FOXM1* promotes the transcription of genes such as *c-myc* and *CCND1*. EMT is promoted and stem cell features are acquired, resulting in an aggressive phenotype (Hyde phenotype). *FOXM1* is likely to enhance GLUT1 expression. (**Right panel**) HCC cells with a high *HNF4α* status. β-catenin translocated into the nucleus promotes the transcription of *HNF4α*, represses the transcription of Snail1/2, an EMT-related gene, and increases β-catenin localization to the cell membrane, resulting in a non-aggressive phenotype (Jekyll phenotype). *HNF4α* increases OATP1B3 transporter expression.

**Figure 4 cancers-14-00444-f004:**
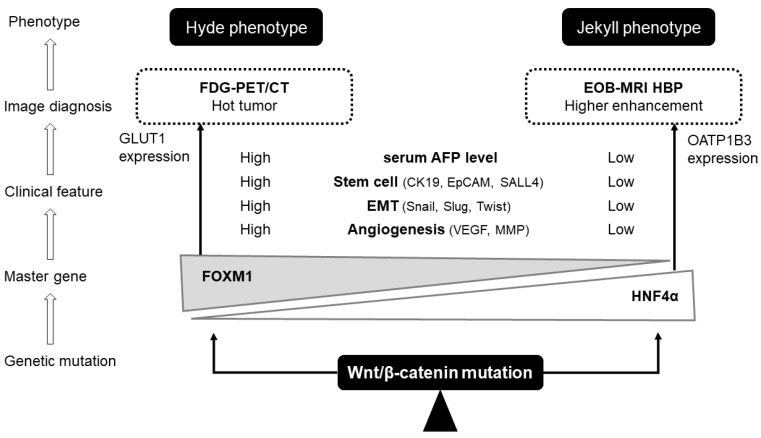
Non-invasive detection of Wnt/β-catenin activation using imaging biomarkers. *HNF4α* and *FOXM1*, which are located downstream of the Wnt/β-catenin canonical pathway, define the phenotype of HCC cells. The Jekyll phenotype with well differentiation and suppressed metastasis and vascular invasion is strongly associated with high expression of *HNF4α*. *HNF4α* promotes the expression of OATP1B3, and we can recognize as iso to higher enhancement intrahepatic nodules in the hepatobiliary phase of Gd-EOB-DTPA-enhanced MRI. The Hyde phenotype with poorly differentiation and massive metastasis and vascular invasion, is strongly associated with high expression of *FOXM1*. *FOXM1* promotes the expression of GLUT1 and may be recognized as a hot nodule in FDG-PET/CT images.

**Figure 5 cancers-14-00444-f005:**
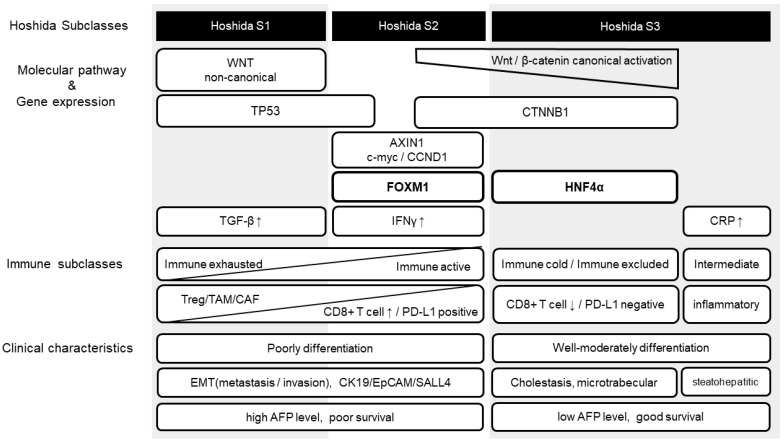
Comparison with previous subclasses. Hoshida subclass 1 is characterized by an enhanced noncanonical WNT/TGF-β axis, resulting in immune hot tumors enriched in tumor-infiltrating T lymphocytes, but also causing T-cell exhaustion in some cases. Hoshida subclass 2 HCCs present *c-myc* and *CCND1* mutations, high serum AFP levels, stem cell features, and enhanced EMT. *FOXM1* expression is upregulated in this subclass, and we believe that some of the canonical Wnt/β-catenin activated HCCs may be herein included. This is probably an immune active and exhausted subclass. A part of Hoshida subclass 3 HCCs presents a good prognosis with *CTNNB1* mutation (activated canonical Wnt/β-catenin pathway), low serum AFP levels, and are well-to-moderately differentiated and non-aggressive. Being an immune cold tumor with fewer tumor-infiltrating T-lymphocytes, it may be difficult to treat, even with immune checkpoint inhibitors.

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
