# Peer review of "Clinical Significance of the Duality of Wnt/β-Catenin Signaling in Human Hepatocellular Carcinoma"

_cancers, 2022, doi:10.3390/cancers14020444_

Round 1
Reviewer 1 Report
This review dealing with HCC by Aoki et al. is quite of interest and very well construted. Schematic respresentations are moreover very clear and help the readers in understanding the text. Here are my comments:
- Ligne 240: authors should add this citation about HCC cells that have increased Epcam expression: Lacoste B, Raymond VA, Cassim S, Lapierre P, Bilodeau M. Highly tumorigenic hepatocellular carcinoma cell line with cancer stem cell-like properties. PLoS One. 2017 Feb 2;12(2):e0171215. doi: 10.1371/journal.pone.0171215. PMID: 28152020; PMCID: PMC5289561.
- Is Figure 3 finished ? It seems by looking at it incomplete due to the other cell membrane that appears on the right ...
- Figure 4 is covering some text - should be modified and corrected.
- Authors mentioned about the Hyde phenotype an increased GLUT-1 expression - they should then definitely describe the Warburg effet displayed by some HCC cells and the increased tumorigenicity associated - metabolic rewiring. To do so, please refer and cite:
- Cassim S, Raymond VA, Dehbidi-Assadzadeh L, Lapierre P, Bilodeau M. Metabolic reprogramming enables hepatocarcinoma cells to efficiently adapt and survive to a nutrient-restricted microenvironment. Cell Cycle. 2018;17(7):903-916. doi: 10.1080/15384101.2018.1460023. Epub 2018 May 21. PMID: 29633904; PMCID: PMC6056217.
- Iansante V, Choy PM, Fung SW, Liu Y, Chai JG, Dyson J, Del Rio A, D'Santos C, Williams R, Chokshi S, Anders RA, Bubici C, Papa S. PARP14 promotes the Warburg effect in hepatocellular carcinoma by inhibiting JNK1-dependent PKM2 phosphorylation and activation. Nat Commun. 2015 Aug 10;6:7882. doi: 10.1038/ncomms8882. PMID: 26258887; PMCID: PMC4918319.
- Liberti MV, Locasale JW. The Warburg Effect: How Does it Benefit Cancer Cells? Trends Biochem Sci. 2016 Mar;41(3):211-218. doi: 10.1016/j.tibs.2015.12.001. Epub 2016 Jan 5. Erratum in: Trends Biochem Sci. 2016 Mar;41(3):287. Erratum in: Trends Biochem Sci. 2016 Mar;41(3):287. PMID: 26778478; PMCID: PMC4783224.
Author Response
Please find attached a revised version of our manuscript, ” Clinical Significance of the Duality of Wnt/β-catenin Signaling in Human Hepatocellular Carcinoma” by Aoki, et al. We would like to thank the reviewers for providing valuable comments on our manuscript to improve its quality. The suggestions given by the reviewers were very helpful in greatly improving the dissertation.
I've rewritten some of the sections you gave me in the attached image to reduce the overlap.
Comments and Suggestions for Authors
This review dealing with HCC by Aoki et al. is quite of interest and very well construted. Schematic respresentations are moreover very clear and help the readers in understanding the text. Here are my comments:
Ligne 240: authors should add this citation about HCC cells that have increased Epcam expression: Lacoste B, Raymond VA, Cassim S, Lapierre P, Bilodeau M. Highly tumorigenic hepatocellular carcinoma cell line with cancer stem cell-like properties. PLoS One. 2017 Feb 2;12(2):e0171215. doi: 10.1371/journal.pone.0171215. PMID: 28152020
Answer:
Thank you for sharing papers with us. I have cited the references.
Is Figure 3 finished ? It seems by looking at it incomplete due to the other cell membrane that appears on the right ...
Answer:
I apologize for the confusing diagram. 
The semicircular cylinder on the right side is a blood vessel adjacent to the HCC cell. It seems confusing, so I will delete it.
Figure 4 is covering some text - should be modified and corrected.
Answer:
Thank you for your suggestion.
I have corrected the text in Figure 4 as follows.
HNF4α and FOXM1, which are located downstream of the Wnt/β-catenin canonical pathway, define the phenotype of HCC cells. The Jekyll phenotype with well differentiation and sup-pressed metastasis and vascular invasion is strongly associated with high expression of HNF4α. HNF4α promotes the expression of OATP1B3, and we can recognize as iso to higher enhance-ment intrahepatic nodules in the hepatobiliary phase of Gd-EOB-DTPA-enhanced MRI. The Hyde phenotype with poorly differentiation and massive metastasis and vascular invasion, is strongly associated with high expression of FOXM1. FOXM1 promotes the expression of GLUT1, and may be recognized as a hot nodule in FDG-PET/CT images.
Authors mentioned about the Hyde phenotype an increased GLUT-1 expression - they should then definitely describe the Warburg effect displayed by some HCC cells and the increased tumorigenicity associated - metabolic rewiring. To do so, please refer and cite:
Cassim S, Raymond VA, Dehbidi-Assadzadeh L, Lapierre P, Bilodeau M. Metabolic reprogramming enables hepatocarcinoma cells to efficiently adapt and survive to a nutrient-restricted microenvironment. Cell Cycle. 2018;17(7):903-916. doi: 10.1080/15384101.2018.1460023. Epub 2018 May 21. PMID: 29633904; PMCID: PMC6056217.
Iansante V, Choy PM, Fung SW, Liu Y, Chai JG, Dyson J, Del Rio A, D'Santos C, Williams R, Chokshi S, Anders RA, Bubici C, Papa S. PARP14 promotes the Warburg effect in hepatocellular carcinoma by inhibiting JNK1-dependent PKM2 phosphorylation and activation. Nat Commun. 2015 Aug 10;6:7882. doi: 10.1038/ncomms8882. PMID: 26258887; PMCID: PMC4918319.
Liberti MV, Locasale JW. The Warburg Effect: How Does it Benefit Cancer Cells? Trends Biochem Sci. 2016 Mar;41(3):211-218. doi: 10.1016/j.tibs.2015.12.001. Epub 2016 Jan 5. Erratum in: Trends Biochem Sci. 2016 Mar;41(3):287. Erratum in: Trends Biochem Sci. 2016 Mar;41(3):287. PMID: 26778478; PMCID: PMC4783224.
Answer:
Thank you very much for your valuable comments.
Based on the papers you provided, I have added a section on glucose metabolism and the Warburg effect in 7.4 to Line 393-Line 408.

Reviewer 2 Report
This is a very well-written review of the dual actions of Wnt/beta-catenin signaling in the development and phenotype of HCC. The authors demonstrate a fine overview of the complex field of HCC development and relate this to the clinical situation. This review can be recommended to clinical hepatologists as well.
Author Response
Please find attached a revised version of our manuscript, ” Clinical Significance of the Duality of Wnt/β-catenin Signaling in Human Hepatocellular Carcinoma” by Aoki, et al. We would like to thank the reviewers for providing valuable comments on our manuscript to improve its quality. The suggestions given by the reviewers were very helpful in greatly improving the dissertation.
I've rewritten some of the sections you gave me in the attached image to reduce the overlap.
Answer:
Thank you very much for taking time out of your busy schedule for peer review.
I'm also very glad for the positive response.

Reviewer 3 Report
Dear Authors,
The manuscript which is entitled with 'Clinical Significance of the Duality of Wnt/β-catenin Signaling in Human Hepatocellular Carcinoma' is very impressive and provide critical issue in the filed of liver study.
To accept 'Cancer' Journal, authors would be needed to add more information to the current form of manuscript.
- Please provide the opinion of micro environment status of immune cells and Wnt/B-catenin Signaling pathway? Authors' individual opinion would be OK.
- Please mention the heterogeneity of HCC and the role of Wnt/B-signaling pathway.
- Please describe the non-canonical pathway and canonical pathway as well in each types.
- Please mention about ligands to each of specific on your manuscript.
Author Response
Please find attached a revised version of our manuscript, ” Clinical Significance of the Duality of Wnt/β-catenin Signaling in Human Hepatocellular Carcinoma” by Aoki, et al. We would like to thank the reviewers for providing valuable comments on our manuscript to improve its quality. The suggestions given by the reviewers were very helpful in greatly improving the dissertation.
I've rewritten some of the sections you gave me in the attached image to reduce the overlap.
The manuscript which is entitled with 'Clinical Significance of the Duality of Wnt/β-catenin Signaling in Human Hepatocellular Carcinoma' is very impressive and provide critical issue in the filed of liver study.
To accept 'Cancer' Journal, authors would be needed to add more information to the current form of manuscript.
Please provide the opinion of micro environment status of immune cells and Wnt/B-catenin Signaling pathway? Authors' individual opinion would be OK.
Please mention the heterogeneity of HCC and the role of Wnt/B-signaling pathway.
Answer:
Thank you for your valuable suggestions.
We have added to the Simple summary and Lines 447-450 and 455-458 about the immune microenvironment in two phenotypes with Wnt/β-catenin activating mutations and heterogeneity of HCCs.
Also, Figure 5 has been rewritten.
Please describe the non-canonical pathway and canonical pathway as well in each types. Please mention about ligands to each of specific on your manuscript.
Answer:
The non-canonical WNT activation pathway corresponds to Hoshida S1, an HCC subclass in which Wnt/TGF-β is activated in an Immune exhausted HCC microenvironment. The canonical WNT pathway is related to Hoshida S2 and S3, and the Jekyll phenotype with high HNF4α expression corresponds to Hoshida S3, and the Hyde phenotype with high FOXM1 expression corresponds to Hoshida S2. I explained the difference between non-canonical pathway and canonical pathway by modifying Figure 5 and other sentences.
Ligands for canonical Wnt pathway were added to Line 133 and Line 439, and ligands for non-canonical pathway were added to Line 468.
Thank you for taking time to review our manuscript. We look forward to hearing from your positive response.
